# Influence of Loading Conditions on the Mechanical Performance of Multifilament Coreless UHMWPE Sutures Used in Orthopaedic Surgery

**DOI:** 10.3390/ma15072573

**Published:** 2022-03-31

**Authors:** Maria Prado-Novoa, Laura Perez-Sanchez, Belen Estebanez, Salvador Moreno-Vegas, Ana Perez-Blanca

**Affiliations:** 1Clinical Biomechanics Laboratory of Andalusia, University of Malaga, Calle Dr. Ortiz Ramos s/n, 29071 Malaga, Spain; lauraperezsanchez@uma.es (L.P.-S.); belen@uma.es (B.E.); smorenov@uma.es (S.M.-V.); anaperez@uma.es (A.P.-B.); 2Telecomunication Research Institute (TELMA), University of Malaga, E.T.S. Ingenieria de Telecomunicaciones, Bulevar Louis Pasteur 35, 29010 Malaga, Spain; 3Biomedical Research Institute of Malaga, Calle Dr. Miguel Díaz Recio, 28, 29010 Malaga, Spain

**Keywords:** UHMWPE sutures, surgical sutures, mechanical properties, orthopaedic surgery, loading conditions

## Abstract

This work studies the influence of loading velocity and previous cyclic loading history on the stiffness and strength of a multifilament coreless ultra-high-molecular-weight polyethylene (UHMWPE) surgical suture. Thread samples (*n* = 8) were subjected to a load-to-failure test at 0.1, 0.5, 1, 5, and 10 mm/s without previous loading history and after 10 cycles of loading at 1–10 N, 1–30 N, and 1–50 N. The experimental data were fitted to mathematical models to compute the stress–strain relation and the strength of the suture. The bilinear model involving two stress–strain ratios for low- and high-strain intervals was the best fit. The ratio in the low-strain range rose with loading speed, showing mean increases of 5.9%, 6.5%, 7.9%, and 7.3% between successive loading speeds. Without a previous loading history, this ratio was less than half than that at high strain. However, 10 cycles of 1–30 N or 1–50 N significantly increased the stress–strain ratio at a low strain level by 135% and 228%, respectively. The effect persisted after 2 min but vanished after 24 h. No influence was found on the suture strength. In conclusion, the stiffness of the studied suture was influenced by the strain level, loading velocity, and recent cyclic loading history. Conversely, the suture strength was not affected.

## 1. Introduction

UHMWPE sutures are a non-resorbable option currently chosen by many orthopaedic doctors for application in surgical repairs of torn soft tissues, i.e., ligaments, tendons, or menisci, or to fix artificial or allogenic implants to the surrounding tissue (e.g., meniscal implants). The use of non-resorbable sutures has been recommended in these surgeries with relatively long post-operative periods [1] due to the lack of a long-term fixation capacity observed in resorbable threads [2,3]. The selection of UHMWPE sutures responds to the improved mechanical properties reported for the suture material [2,4,5,6,7] compared to previously available non-resorbable suturing elements (made of polyamide, polypropylene, or polyester), since the success of such repairs relies on the capacity of the sutures to hold the wounded ends together or to fix tissue in place in order to promote the healing process [8,9]. UHMWPE is a biomaterial also used in other clinical applications, for example, in total hip arthroplasty implants, due to its higher wear resistance compared to other polymers, a complication associated with this surgery that has been studied in the scientific literature [10] and that can be related to the stresses generated in the joint during use [11].

The mechanical properties of sutures may include those of the suture material [4,5,6,12], suture knots [13,14] and the suture–tissue interface [15,16]. Focusing on the suture material, the tensile strength parameter related to the capacity to withstand loads before failure is the only mechanical property standardised for sutures in the United States Pharmacopeia (USP) [9] or European Pharmacopeia [17], and it is the most widely reported mechanical property in the scientific literature. Stiffness, a parameter characterising thread elongation, is also frequently computed as it is related to the ability of the thread to maintain enough tissue approximation for proper healing under tensile loads; however, an overly stiff suture material will be difficult to work with when performing the knot due to its low ability to comply with the topology of the surrounding tissues [18]. In orthopaedic surgery, among other characteristics, the ideal suture should have enough resistance to withstand the loads during the intervention and in the early post-operative period when the fixation relies on the suture since the biological process of healing that leads to tissue generation has not yet begun. In addition, during this period, it is even more important that suture stiffness is adapted to the repaired tissue in order to avoid cutting through it, while it is high enough to prevent gap formation.

Following orthopaedic surgery, it is common to mobilise the repaired area in rehabilitation protocols that subject the sutures to loads applied at different speeds and normally in repetitive routines. In this context, knowledge of the viscoelastic response of sutures is necessary in order to predict its influence in the progression of healing.

Prior studies assessed the mechanical properties of different commercially available UHMWPE sutures. Following United States Pharmacopoeia standards, the ultimate failure loads were determined in load-to-failure tests conducted at single-loading velocities [4,5,6,12]. The knot security of different knots was analysed in terms of knot strength and slipping resistance measured as plastic deformation after cyclic tests at normal physiological load levels [13,14,19,20]. In addition, viscoelastic static and dynamic creeps were characterised for suture strands alone [21,22] or tied with knots [15] in tests that, again, were conducted at single loading velocities.

All this previous research provides valuable information on the mechanical behaviour of ultra-resistant polyethylene sutures, though it is far from being complete. The purpose of this work was to study three characteristics of braided multifilament coreless UHMWPE surgical sutures that may have a direct impact on the definition of the rehabilitation protocol after surgery in order to ensure the success of the healing process. They have not been quantified to date and are as follows:The non-linearity of the stiffness which would affect the elongation at different load levels.The dependence of the resistance and stiffness on the loading velocity which would limit the speed and intensity of the exercises.The influence of the previous cyclic loading and resting periods on single-loading properties.

## 2. Materials and Methods

In this study, the N°2 ForceFiber^®^ white suture thread (Teleflex, ResearchTriangle Park, NC, USA, marketed by Stryker Corporation) was tested. It was chosen as a representative example of an ultra-resistant polyethylene suture with extended use among orthopaedic surgeons. The selected multifilament thread has a coreless braided configuration of an undyed 100% UHMWPE fibre. According to the USP [9], the cross-sectional diameter of a N°2 synthetic suture is 0.5495 (±0.0495) mm, which results in an apparent area of 0.2372 (± 0.0019) mm^2^ assuming a circular cross-section. Suture samples of approximately 150 mm were cut and randomly assigned to the 10 study groups described in Section 2.1, Section 2.2 and Section 2.3. The group size was *n* = 8 for all studies.

A uniaxial traction/compression, custom-designed by Bioclina (Clinical Biomechanics Laboratory of Andalusia, Malaga, Spain) for biomechanical applications, was used for the tests (Figure 1). The loading system consists of a vertical ball screw (Accuslide 2HBE20, Tecnopower, Barcelona, Spain) powered by an electrical servomotor (Servomotor Sigma II, Omron Electronics Iberia, S.A.U., Madrid, Spain). The central processing unit (CPU) is an embedded device (NI myRIO, National Instruments) that includes analogue inputs and digital outputs and a dual-core ARM Cortex-A9 processor. To align the sample with the traction direction, the machine features two screw grips: one attached to the actuator and the other fixed to the base of the machine by means of a clamp with 5 degrees of freedom (TLT/SP-75 model, Wilton Tools, La Vergne, TN, USA). This does not allow for displacement in the direction of the load which is executed by the machine actuator. The inner surfaces of both clamps were covered with sand paper to increase sliding resistance. The load is measured by a uniaxial 0.1 load cell of 2 KN nominal rating (HBM, Darmstadt, Germany) located on the machine head. The relative position between both grips is gauged by a linear variable differential transformer (LVDT) with a 0.03 mm resolution (LVP-100-ZA-2.5-SR7-I, Micro-Epsilon, Ortenburg, Germany). The signals can be sampled up to 1000 Hz and are recorded in two independent binary files that include the time reference common to both sensors. The binary files were imported in MATLAB R2020b (Mathworks Inc., Natick, MA, USA). The same software was used to perform the calculations described below (excluding the statistical analyses described in Section 2.5). Possible test setup inaccuracies due to small thread slippage or border effects at the gripping points were discarded in previous pilot tests (see Appendix A); nonetheless, to check for any macro slippage of the sample at the grips, it was marked with a straight line at the exit of each one. Initial suture length between the gripping points was set at 80 mm.

From LVDT recordings, the stretch ratio, λ, was computed as
(1)$$\lambda =\frac{L}{L_{0}}$$
where $L_{0}$ refers to the suture length at the test initiation. The engineering stress, σ, was calculated as
(2)$$\sigma =\frac{F}{A}$$
where F is the traction force registered by the load cell, and A is the mean unloaded apparent cross-section of N° 2 sutures.

### 2.1. Proposed σ−λ Curve–Fitting Models

The mechanical performance of the UHMWPE sutures was analysed by proving the fit of the experimental data against three sets of curves used to describe different material behaviours. The curves studied to model σ as a function of λ were as follows:Linear model:
(3)$$\sigma =p_{1}\lambda +p_{2}$$

Bilinear model:


(4)
$$\begin{array}{cc} \sigma =p_{11}\lambda +p_{21} & for\;1\leq \lambda \leq \lambda_{trn}\\ \sigma =p_{12}\lambda +p_{22} & for\;\lambda_{trn}<\lambda \;\;\;\;\;\;\;\; \end{array}$$



where $\lambda_{trn}$ is a transition point that divides the range into two strain intervals which we referred to as low and high strain levels. It is a 5-parameter model, since $\lambda_{trn}$ is searched during the fitting process along with *p*_11_, *p*_21_, *p*_12_, and *p*_22_.

A 3-parameter polynomial model for hyperelastic incompressible materials known as the Signiorini model [23]:


(5)
$$\sigma =2C_{10}\left(\lambda -\frac{1}{\lambda^{2}}\right)+2C_{01}\left(1-\frac{1}{\lambda^{3}}\right)+4C_{20}\left(\lambda -\frac{1}{\lambda^{2}}\right)\left(\lambda^{2}+\frac{2}{\lambda }-3\right)$$


The parameters of the models in Equations (3)–(5) were searched to minimise the root mean square error (RMSE) between the function and experimental $\left(\sigma ,\;\lambda \right)$ points.

The fitting process was performed on data from groups of samples subjected to displacement-controlled load-to-failure tests at different loading velocities. Specifically, the samples were loaded with a 5 N traction force for 10 s followed by a load-to-failure test conducted at 5 loading velocities: 0.1 mm/s, 0.5 mm/s, 1 mm/s, 5 mm/s, and 10 mm/s.

The range of loading velocities was selected as representative of human joint movements, from a quasi-static displacement (0.1 mm/s) to maximum velocity in a harmonic cyclic loading with an amplitude of 50 N, which is representative of the peak tension on the repaired meniscal root during open-kinetic-chain exercises against a resistant force of 30 N at the ankle [24,25] as a habitual orthopaedic surgical treatment, performed at a customary frequency of 1 Hz. This implies a displacement from the equilibrium point given by
(6)$$x=\frac{F^{*}}{k}\cos\left(2\pi ft\right)$$
where $F^{*}$ is the loading amplitude, f is the frequency of the exercise, t is the time, and k the stiffness of the suture. The latter was estimated as 30 N/m from the results of our pilot test described in Appendix B (consistent with the results subsequently found in Section 3.1, Section 3.2 and Section 3.3 of this document). The amplitude of the velocity in such exercises was computed as 10.47 mm/s following Equation (7):(7)$$v=\left|\frac{dx}{dt}\right|=\frac{F^{*}}{k}2\pi f$$

The goodness of fit was assessed in terms of the adjusted R-square, which reflects the fit quality of the model; and the RMSE, which reflects the discrepancy between the observed and expected data. Only the interval of λ with data from the 8 samples available at all velocities was used, so the fitting interval was upper bounded by the maximum λ value of the sample that failed at the lowest stretch ratio.

### 2.2. Effect of Loading Velocity on the Suture Resistance

To evaluate whether the strength and flexibility of the sutures was conditioned by the loading velocity, the outcomes of the 5 groups of the load-to-failure tests described in Section 2.1 were analysed.

Suture resistance was studied based on the ultimate stress, $\sigma^{ult}$, as the maximum stress of the load-to-failure test; and the ultimate stretch ratio, $\lambda^{ult}$, as the stretch-ratio at which the maximum stress point occurs. Regarding stiffness, defined as the slope of σ−λ, it was calculated from the models fitted as follows:Linear model of Equation (3):
(8)$$E=p_{1}$$Bilinear model of Equation (4), two different values for two strain levels, low and high:
(9)$$\begin{array}{cc} E^{low}=p_{11} & for\;1\leq \lambda \leq \lambda_{trn}\\ E^{high}=p_{12} & for\;\lambda_{trn}<\lambda \;\;\;\;\;\;\;\; \end{array}$$The hyperelastic model from Equation (5), a parameter continuously dependent on the stretch ratio:
(10)$$E=\frac{d\sigma }{d\lambda }$$
E in Equations (8)–(10) is also the stress–strain rate of the sutures, since the strain (ε) and the stretch ratio are related by
(11)ε=λ−1

### 2.3. Effect of a Previous Cyclic Loading on Suture Stiffness

Load-to-failure tests at approximately 1 mm/s (selected as a common loading velocity in testing surgical sutures [12,26,27])—preceded by a loading history consisting of 10 cycles at subcritical loads and 1 Hz—were planned to assess whether suture stiffness and suture strength are modified by cyclic loading. Ten cycles were considered sufficient, after verifying in previous tests that the greatest influence of repetitive loading on suture stiffness is evident in these first loading cycles (see Appendix C). The cyclic loading was conducted at three subcritical load ranges, 1–10 N, 1–30 N, and 1–50 N, to check the possible influence of the magnitude of a previous load on the mechanical properties of the sutures.

Ultimate stress, $\sigma^{ult}$, and the corresponding ultimate stretch ratio, $\lambda^{ult}$, as defined in Section 2.2, were computed for the 3 groups of load-to-failure tests performed after each cyclic load range. The σ−λ curves were adjusted to the bilinear function given in (4), since it was the function that best fit to the curve (see Section 3.1). As in the previous section, the upper limit of the stretch ratio interval was selected to have data from the 8 samples at every instant. Finally, the stress–strain rate for low and high strain levels according to (9) was calculated.

### 2.4. Persistence over Time of the Effects of Previous Cyclic Loading

The recovery of the mechanical properties after unloading was studied. The test described in Section 2.3 was repeated at the cyclic load range 1–50 N, but a relaxation period of the suture was programmed before the load-to-failure test. Two resting periods were compared: a reduced one of 2 min and a much longer one of 24 h period. The ultimate strain ($\varepsilon^{ult}$), $\lambda^{ult}$, and stress–strain rate at low and high strain levels were computed as in previous section (see Equation (9)).

The most demanding cyclic load range, 1–50 N, was selected for the study after it was verified as the one that yielded the greatest changes in the mechanical properties of the suture among those tested (see Section 3.3).

### 2.5. Statistical Analysis

A priori power analysis was performed using G-Power* 3.1.9.7 [28]. Assuming linear behaviour, a minimum sample size of 7 specimens per group was necessary to find differences in stiffness with loading velocity for a one-way analysis of variance (ANOVA) test at α = 0.05, 1 − β = 0.95, and an effect size of f = 0.8, computed based on the first three specimens of each group. From this estimation, an equal group size of *n* = 8 was chosen, which is in line with the number of specimens tested in previous experimental studies of ultra-resistant UHMWPE suture materials [6,12,21,22].

Statistical analysis was performed using the statistical software package IBM SPSS Statistics, v 25 (IBM, Chicago, IL, USA). Mean values and standard deviations (SD) of the study variables were calculated for every group. In each study, after all variables passed the Shapiro–Wilk test of normality, one-way ANOVA tests were conducted to evaluate differences between groups. In the study on the influence of loading velocity, for the variables that showed significant differences, the type of dependency with velocity was analysed using a post hoc polynomial trend contrast. In the remaining studies, where applicable, differences between groups were studied conducting post hoc pairwise comparisons with Bonferroni or Games–Howell adjustments depending on whether the variables passed the Levene’s equality of variances test. *p*-values < 0.05 were regarded as significant and all tests were 2-tailed. When comparing the parameters obtained by fitting experimental data to different mathematical models at each loading velocity tested, two-tailed paired samples t-tests were conducted.

## 3. Results

### 3.1. Modelling of the σ−λ Curve

Figure 2a displays the $\left(\sigma ,\;\lambda \right)$ data points of the load-to-failure tests with no previous loading history for the five testing groups. The data at each loading velocity were fitted by least squares method to the models in Equations (3)–(5) within the interval of 1≤λ≤1.077. The stretch states are upper bounded by the minimum ultimate strain of the 8 × 5 samples. Table 1 shows the adjusted R-square and the RMSE achieved with the three mathematical models and Figure 2b–d show the fitted models.

The results reveal a non-linear mechanical behaviour of the sutures with an important variation in the stiffness with the strain level. As shown in Table 1, the best fit, in terms of minimum RMSE and maximum R-square, was found by the bilinear model for all velocities (Figure 2c). Thus, the dependency of stiffness on stretch could be correctly modelled by two values for low and high strain levels. The transition points which minimised the errors of the bilinear fit were found for all the specimens close to the midpoint of the interval ($\lambda_{trn\;}$ = 1.038 (0.003) and mean (SD) pooled for the five testing groups).

The hyperplastic model (Figure 2d) also provided a good modelling of the suture behaviour, although with moderately lower goodness and a particularly worse fit in the representation of the low stress range (below 150 MPa), which is more interesting for clinical applications. Regarding the linear model, although it provided an acceptable goodness of fit, the computation of the stress–strain ratio by Equation (3) resulted in values of E significantly different to those found by the bilinear model. Depending on the loading speed, $E^{low}$ was between 49% and 63% lower than E (*p* < 0.001 for all groups), while $E^{high}$ was between 26% and 33% higher (*p* < 0.001 for all groups) (see Appendix D)_._

### 3.2. Influence of the Loading Velocity

Analysing the outcomes of the bilinear model, i.e., the best fit found in the previous section, two different stress–strain rates were computed for low and high strain levels, and they were significantly different (*p* < 0.001 for all the groups), $E^{high}$ being more than double $E^{low}$ for all loading velocities (Table 2). A continuous rise in the stress–strain rate with loading speed was observed in the low strain range (a mean increase in $E^{low}$ of 5.9% from 0.1 to 0.5 mm/s, 6.5% from 0.5 to 1 mm/s, 7.9% from 1 to 5 mm/s, and 7.3% from 5 to 10 mm/s) which was not found for $E^{high}$. Indeed, the ANOVA test revealed significant differences between groups for $E^{low}$ (*p* = 0.028) but not for $E^{high}$ (*p* = 0.420). A post hoc polynomial trend contrast resulted in a significant linear dependency of the stress–strain ratio on the low strain level with the loading velocity (*p* = 0.002). Differences were also found by the ANOVA test for the load at the transition point, $F_{trn}$ (*p* = 0.011), but not for $\lambda_{trn\;}$ (*p* = 0.166). This outcome was explained by the increase in $E^{low}$ as the loading velocity rises, which implies that a higher tensile force causes similar stretching as the loading rate increases.

As for suture strength, no noticeable impact attributable to loading speed was found in the ultimate stress or ultimate stretch ratio according to the values shown in Table 2 for these parameters.

### 3.3. Influence of a Previous Cyclic Loading

Figure 3 displays the σ−λ plot based on the experimental data recorded during the load-to-failure tests at 1 mm/s performed without previous cyclic load and after 10 loading cycles at 3 levels (1–10 N, 1–30 N and 1–50 N). Continuous lines show the bilinear fits of (4). Parameters related to suture mechanical performance computed after the fitting processes are given in Table 3.

In the tests with no previous cyclic loading, no clear stiffness transition point was found on the σ−λ plot, so the mathematical optimisation in Section 3.1 always placed the transition point of Equation (3) in the middle of the interval 1≤λ≤1.077. A similar behaviour was found for the group with the lowest cyclic preload of 1–10 N, for which $\lambda_{trn}$ was also found in the middle of the optimisation interval. However, for the groups with the highest levels of prior cyclic loading (1–30 N and 1–50 N), a point of slope change was clearly perceived. As a result, the $\lambda_{trn}$ values of these two groups were significantly different from both the group with no load history (*p* = 0.008 for the 1–30 N group, *p* = 0.002 for 1–50 N) and the group with the lowest cyclic loading (*p* < 0.001 for the 1–30 N group, *p* < 0.001 for 1–50 N).

$E^{low}$ was also significantly different among the distinct previous cyclic loads (*p* < 0.001 for 1–10 N vs. 1–30 N, 1–10 N vs. 1–50 N, and 1–30 N vs. 1–50 N) and increased with the preload level. Compared to the direct pull-out without preload, the increase in mean $E^{low}$ was 5% for the cyclic preload level 1–10 N, 135% for 1–30 N, and 228% for 1–50 N, only reaching significance for the higher levels (*p* < 0.001 for 1–30 N and *p* < 0.001 for 1–50 N). $E^{high}$ was much less altered by preload compared to direct pull-out, with no significant differences found for any level. Additionally, it was observed that $E^{high}$ was higher than $E^{low}$ for the groups with the lowest level or no previous cyclic load, while the opposite was found after the higher-level pre-loads. This finding was due to the important rise of $E^{low}$ yielded by the most demanding cyclic preload levels since differences in $E^{high}$ were not significant.

### 3.4. Time Persistence of Previous Cyclic Loading Effects

Figure 4 displays the experimental records when the load-to-failure test was performed after different relaxation periods (non-stop, 2 min or 24 h) following the cyclic loading at level 1–50 N. Results with no prior cyclic loading are also included in the figure for comparative purposes. Mechanical performance of the sutures computed from the test after 2 min of rest had mean values similar to those from the test immediately carried out after the repetitive loads (no significant differences for any variable in Table 4). As a result, the variables affected by cyclic loading in the test with no rest also showed significant differences for the group with 2 min rest from the group with no loading history (*p* < 0.001 for $E^{low}$, *p* = 0.001 for $E^{high}$, *p* < 0.001 for $\lambda_{trn}$). Conversely, the test performed after a rest of 24 h showed similar results to the test with no prior cyclic loading (no significant differences for any variable in Table 4), suggesting that, 24 h later, the influence of the loading history had vanished.

As expected after the study in Section 3.3, no difference was found between groups regarding strength.

## 4. Discussion

The main findings of our work were that the stiffness of the studied coreless UHMWPE surgical suture was influenced by the strain level, loading velocity, and previous cyclic loading history, whereas the alteration produced by a loading history is not permanent over time. Conversely, the suture strength was not affected by these parameters.

It was found that the stress–strain curve of the suture thread was adequately modelled by a bilinear function within the stretch ratio range 1≤λ≤1.077. A significantly different stiffness was shown at the low- and high-strain-level intervals resulting from the bilinear modelling of the σ−λ curve. The maximum traction force subjected by the samples in the aforementioned range was 146.14 N on average (SD 29.98); thus, the fitted bilinear model was suitable for a range that covers the traction forces expected in the immediate postoperative period of orthopaedic surgeries [25,29,30,31], particularly if loading is restricted immediately after surgery—as it usually is—and assuming that habitual orthopaedic surgical techniques plan for at least two stitches.

In the load-to-failure tests performed without prior cyclic loading, the stress–strain rate computed by the bilinear model for the low strain range, $E^{low}$, was less than half of that for the high strain range, $E^{high}$. $E^{low}$ was between 49% and 63% higher, and $E^{high}$ was between 26% and 33% lower compared with the value estimated by the linear model in interval 1≤λ≤1.077. Therefore, a linear model used to compute suture stiffness [4,26,27] should be restricted to the use of data in the strain range that is expected for its subsequent clinical use.

A rise in the loading velocity of the sutures in the interval between 0.1 mm/s and 10 mm/s caused a continuous increase in the stress–strain rate in the low strain range, with no differences found in the high strain range. This finding highlights the importance of choosing a testing velocity adjusted to the specific clinical use of the sutures when characterising the elasticity of UHMWPE sutures. This approach is not customary in other published studies, and is not even considered in the Standard Test Method for Tensile Properties of Yarns by the Single-Strand Method [32], which sets a single velocity for displacement-controlled load-to-failure tests not selected for any specific clinical applicability. This rate is not consistent among authors [4,12,26,27] although values at approximately 1 mm/s are common. This consideration may not be pertinent when evaluating the strength of suture, since this is not influenced by loading speed in the studied range, which comprises the values expected during the immediate postoperative period of orthopaedic surgeries. To the best of our knowledge, this was the first work to report this observation.

We performed load-to-failure tests immediately after 10 cycles of low (1–10 N), medium (1–30 N), and high (1–50 N) subcritical loading. No change was found in the studied mechanical properties of the suture after the less demanding cyclic loads. However, for the two most demanding tests, the shape of the σ−λ curve changed. On the one hand, the bilinear model resulted in a low strain range with an upper bound clearly related to the maximum suture stress generated by the previously applied cyclic load. Consequently, the application of $E^{low}$ was reduced to lower strain intervals, but widened towards greater strains as the previous cyclic load increased. It did not reach the extension of the interval without previous loading for the studied values. On the other hand, a marked increase in the stiffness of this low strain range was observed, while the stiffness after the transition point did not show differences compared to straight pull-out; however, as mentioned above, its interval of application was extended towards lower strain values. From a clinical perspective, the mobilisation and exercise regimen in the immediate postoperative period normally impose traction forces on the sutures that must be limited to avoid the formation of gaps at the repair site which could compromise healing. The underestimation of suture stiffness when repetitive loads are applied causes an overestimation of their elongation, which would result in an over-limitation of mobility, and thus, a rehabilitation plan that is less effective in bone remodelling or the maintenance of muscle mass. Therefore, we believe that an increased initial stiffness observed for cyclic forces that exceeds 30 N and modified strain interval limits should be considered when studying rehabilitation protocols.

It is a routine clinical practice to immediately precondition some tissues before their surgical use in order to adapt their behaviour to the loading conditions under which they are expected to work. As the stiffening of the sutures disappeared with rest, the effect of repetitive loads being negligible 24 h later, from our point of view, is that a hypothetical traction preconditioning of the material by the manufacturer would have no effect at the time of use of the suture. While preconditioning by the surgeon immediately before use could have an unexpected influence on the surgical results since the stress–strain ratio of the thread remained modified 2 min after cyclic loading but the effect disappeared with time. Although we consider that the expected loading conditions and level of deformation should be taken into account when selecting a UHMWPE surgical suture, we also considered that such information will not be of interest when the only purpose of a study is to evaluate suture resistance as we have found that the ultimate load is not affected by the loading conditions. This is the case in the Standard Test Method for Tensile Properties of Yarns [32] which proposes a protocol for testing the thread at a single speed.

To take advantage of the improved resistance of the new polyethylene sutures, knotless surgical techniques are arising for which the mechanical performance of the repair depends on the suture thread itself and not on the knot, highlighting the importance of the characterisation of isolated threads. The material strength of commercial braided coreless UHMWPE N.2 suture threads, including HiFi^®^ (ConMEd, Utica, NY, USA), Ultrabraid™ (Smith and Nephew, Andover, MA, USA), HerculineTM (Linvatec, Largo, FL, USA), and ForceFiber^®^ (Teleflex Medical OEM, Tenosha, WI, USA), have previously been reported [4,6,12] in terms of the maximum force experienced before breakage in load-to-failure tests. Mean values were in the range between 213 N [6] and 250 N [12]. The variability is explained in part by differences in cross-section and thread designs, including weave patterns or the diameter of individual fibres [12]. Although these values were obtained at different loading velocities from 20 mm/min [6] to 60 mm/min [4], they can be compared since we found no dependency of the ultimate stress on loading velocity. By pooling the data obtained for direct pull-out at the different velocities tested, we obtained a mean maximum strength of 192.71 N which approximates the range of prior works. On the other hand, Wüst et al., (2006) [4] published values of mean elongations at a maximal load that approximated 32% for Herculine and 36% for Ultrabraid N.2 threads. In our work, by taking great care to ensure that measured elongations did not include any slipping of the thread in the grips, the Force Fiber thread of the same grade demonstrated a substantially superior resistance to elongation at high loads with a pooled mean strain at ultimate load for all loading velocities of 10.2%. The technique used in prior studies to grip the strands by wounding each end to a metallic rod in the testing machine may have contributed to the unexpected large difference found between these otherwise similar thread products.

Despite the recognised clinical implications of suture material stiffness [3,33], the definition of the parameter is not standard and little is found for UHMWPE sutures in the literature. Cardoso et al., (2019) [6] obtained force vs. elongation curves in a tensile test of different suture threads depicting a continuous increase in the stiffness of Hi-Fi^®^ coreless sutures, as opposed to cored braided polyester (Ethybond^®^, Ehicon Inc., Somerville, NJ, USA), which presented a higher initial stiffness; or cored braided UHMWPE (FiberWire^®^, Arthrex, Naples, FL, USA) with an almost linear behaviour until failure. In our study, the ForceFiber suture showed a non-linear σ−λ behaviour similar to Hi-Fi, which could be explained by the coreless design of both threads, although the parameters that characterise each material vary, probably due to differences in testing conditions and thread design.

The importance of viscoelastic effects has long been recognised [34]. However, very few works have addressed the intrinsic viscoelastic properties of UHMWPE sutures [20,21] such as creep and relaxed elongation under static and cyclic loads, comparing diverse commercial sutures under different environmental conditions. In our study, conducted in dry air, we gained valuable information about the influence of the loading velocity, a parameter of acknowledged importance for which we could not find any published information in reference to UHMWPE sutures. In addition, we introduced other unedited parameters related to the viscoelastic nature of braided UHMWPE that influenced single pull-out stiffness and strength, such as a previous history of cyclic subcritical loads at different levels expected in the postoperative period followed by distinct relaxation periods. The results of this work highlight that stiffness characterisation with an isolated direct load-to-failure test at a fixed speed may not provide sufficient data about their performance.

A limitation of this study was the use of a single surgical suture type for all tests. Obviously, the results are not directly transferable to all UHMWPE sutures, as they may consist of pure UHMWPE or be blended with different polymers at different percentages; they may or may not have a core; they may or may not be coated; etc. Even for multifilament coreless braided suture threads similar to those used in this study, differences in the weave pattern, the diameter of the fibres, or the cross-sectional area may exist that prevent the direct extrapolation of the values of properties obtained herein. The cross-sectional area of the sutures was not measured in our study and may have contributed to observed differences in the mechanical properties compared to different sutures. However, ForceFiber^®^ threads come in standard sizes, and our main purpose was to study the differences in the mechanical behaviour of these single coreless ultra-high suture threads with loading conditions. Another limitation refers to the possible changes in the mechanical properties of sutures in vivo when they are immersed in a physiological environment, reported for the viscoelastic parameters by some authors [21,22] but discarded by others [5,35]. Finally, we did not consider possible suture degradation by abrasion or chemical processes, but it should be noted that we studied isolated suture threads of non-resorbable material which are made to retain their principal mechanical properties after 60 days [2,3], a longer time period than the immediate postoperative period, after which healing processes occur and as a result, loads are not exclusively borne by the suture.

## 5. Conclusions and Future Work

The stiffness of braided multifibre coreless 100% UPHWE surgical suture threads is influenced by loading velocity in the range of 1–10 mm/s and by previous cyclic loading where the alteration produced by this loading history is not permanent over time and non-noticeable after 24 h. Conversely, suture strength is not affected by these parameters. The variation of stiffness with the strain level is better represented by a bilinear model of the stress–strain curve than by a linear or a three-parameter hyperelastic model.

Future research is needed to establish the level of cyclic loading which more precisely produces an alteration of the stiffness of the suture and to investigate its possible relationship with the properties of constituent materials and/or the characteristics of the suture associated with the manufacturing process. Similarly, it would be helpful to better define the period of time required for such alterations to dissipate. Finally, it would be necessary to verify whether the mechanical properties of other UHMWPE sutures were also modified by loading conditions.

## Figures and Tables

**Figure 1 materials-15-02573-f001:**
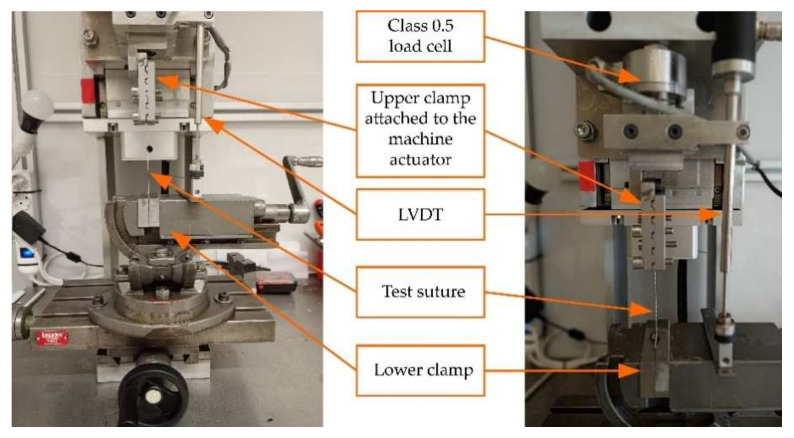
Uniaxial traction/compression testing machine.

**Figure 2 materials-15-02573-f002:**
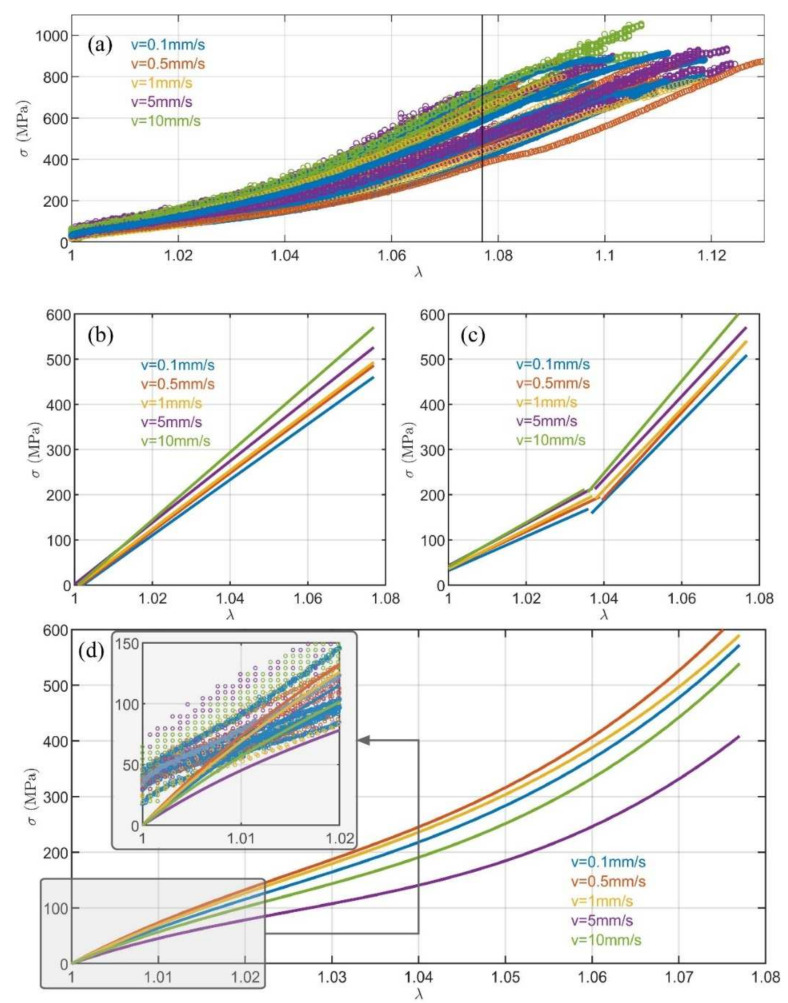
Stress–stretch ratio of the load-to-failure test at 0.1 mm/s, 0.5 mm/s, 1 mm/s, 5 mm/s, and 10 mm/s: (**a**) experimental data points; (**b**) linear fitted models; (**c**) bilinear fitted models; and (**d**) hyperelastic fitted models, with a magnification of interval 1≤λ≤1.02, also showing the markers of the experimental data.

**Figure 3 materials-15-02573-f003:**
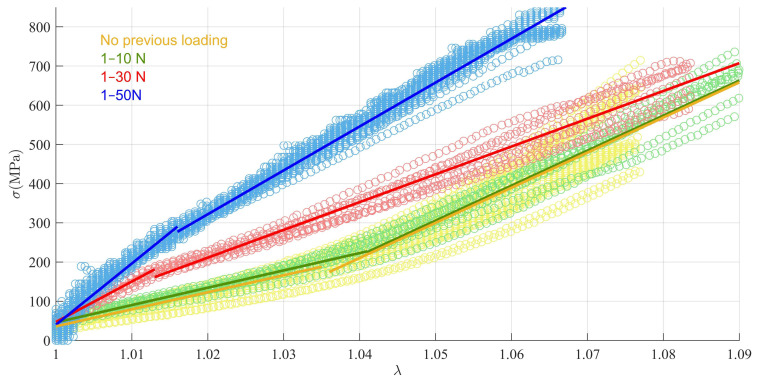
Results of the load-to-failure tests without and immediately after different cyclic loading ranges. Markers indicate the experimental data. Continuous lines are the fitted bilinear models.

**Figure 4 materials-15-02573-f004:**
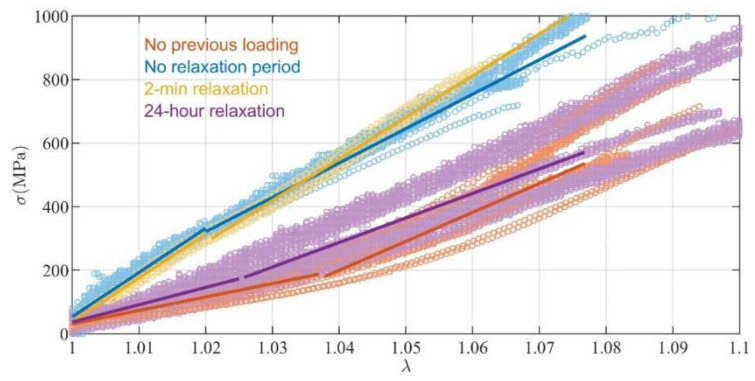
The experimental data of the load-to-failure tests performed without prior cyclic loading; immediately after 10 cycles of loading at 1–50 N; after a relaxation period of 2 min after cyclic loading; and after a relaxation period of 24 h after cyclic loading. Markers indicate the experimental data. Continuous lines are the fitted bilinear models.

**Table 1 materials-15-02573-t001:** Goodness of fit for the mathematical models of Equations (3)–(5) in terms of adjusted R-square and RMSE at each loading velocity in the load-to-failure test. Values are given as mean (SD) for the 8-sample groups.

Model	Goodness	0.1 mm/s	0.5 mm/s	1 mm/s	5 mm/s	10 mm/s
Linear model	*Adj. R*	0.95 (0.01)	0.96 (0.00)	0.96 (0.01)	0.97 (0.01)	0.97 (0.01)
	*RMSE*	29.20 (3.31)	31.46 (7.69)	28.44 (6.06)	26.70 (6.31)	32.11 (2.70)
Bilinear model	*Adj. R*	1.00 (0.00)	1.00 (0.00)	1.00 (0.00)	1.00 (0.00)	1.00 (0.00)
	*RMSE*	7.32 (0.95)	7.03 (1.43)	7.56 (1.44)	6.86 (1.42)	9.51 (2.28)
Hyperelastic model	*Adj. R*	0.99 (0.00)	0.99 (0.00)	0.99 (0.00)	0.990 (0.00)	0.99 (0.00)
	*RMSE*	9.50 (2.81)	11.49 (2.30)	10.35 (3.07)	15.46 (3.14)	15.43 (3.53)

**Table 2 materials-15-02573-t002:** Results of the load-to-failure test fitted by the bilinear model for the 5 groups at different loading velocities. Values are given as the mean (SD).

Loading Velocity(mm/s)	$E^{low}$	$E^{high}$	$\lambda_{trn}$	$F_{trn}$	$\sigma^{ult}$	$\lambda^{ult}$
	(MPa)	(MPa)		(N)	(MPa)	
0.1	3754	8789	1.037	33.20	786.06	1.108
	(904)	(1336)	(0.003)	(4.95)	(79.79)	(0.077)
0.5	3976	9482	1.040	37.87	832.12	1.103
	(665)	(2021)	(0.003)	(3.66)	(47.13)	(0.015)
1	4236	9117	1.038	37.44	723.82	1.095
	(889)	(1753)	(0.003)	(7.31)	(108.97)	(0.015)
5	4570	9207	1.038	42.13	846.13	1.106
	(674)	(1675)	(0.003)	(5.42)	(75.82)	(0.016)
10	4903	10,238	1.036	42.57	874.11	1.098
	(561)	(839)	(0.002)	(6.70)	(124.23)	(0.009)

**Table 3 materials-15-02573-t003:** Parameters of the bilinear model fits the data of load-to-failure tests performed at 1 mm/s with no previous cyclic loading and after 10 loading cycles at 1–10 N, 1–30 N, and 1–50 N. Values are given as mean (SD).

Previous Cyclic Loading	$E^{low}$	$E^{high}$	$\lambda_{trn}$	$F_{trn}$	$\sigma^{ult}$	$\lambda^{ult}$
	(MPa)	(MPa)		(N)	(MPa)	
No cyclic	4236	9117	1.038	37.44	723.82	1.095
	(889)	(1753)	(0.003)	(7.31)	(108.97)	(0.015)
1–10 N	4459	9058	1.042	45.54	881.87	1.116
	(321)	(1038)	(0.004)	(3.20)	(111.96)	(0.016)
1–30 N	9947	7319	1.014	37.73	837.22	1.110
	(754)	(667)	(0.003)	(4.98)	(124.33)	(0.024)
1–50 N	13,908	10,823	1.020	63.69	844.99	1.072
	(871)	(771)	(0.008)	(18.25)	(117.08)	(0.013)

**Table 4 materials-15-02573-t004:** Parameters of the bilinear model fit from the data of the load-to-failure tests performed at 1 mm/s without previous cyclic loading; immediately after 10 loading cycles at 1–50 N; after a relaxation period of 2 min following cyclic loading; and after a relaxation period of 24 h following cyclic loading. Values are given as the mean (SD).

Previous Cyclic Loading	$E^{low}$	$E^{high}$	$\lambda_{trn}$	$F_{trn}$	$\sigma^{ult}$	$\lambda^{ult}$
	(MPa)	(MPa)		(N)	(MPa)	
No cyclic	4236	9117	1.038	37.44	723.82	1.095
	(889)	(1753)	(0.003)	(7.31)	(108.97)	(0.015)
No relax	13,908	10,823	1.020	63.69	844.99	1.072
	(871)	(771)	(0.008)	(18.25)	(117.08)	(0.013)
2 min relax	13,791	13,096	1.021	63.60	792.75	1.060
	(1051)	(433)	(0.003)	(5.92)	(59.24)	(0.005)
24 h relax	5445	7718	1.026	34.00	782.72	1.105
	(834)	(1263)	(0.010)	(6.91)	(115.42)	(0.008)

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
