# Peer review of "Influence of Loading Conditions on the Mechanical Performance of Multifilament Coreless UHMWPE Sutures Used in Orthopaedic Surgery"

_materials, 2022, doi:10.3390/ma15072573_

Round 1
Reviewer 1 Report
Dear Prado-Novoa et al.,
The manuscript “Loading conditions affect the mechanical performance of multifilament coreless UHMWPE sutures used in orthopaedic surgery” (materials-1640517) by Prado-Novoa et al. studies the influence of loading velocity and previous cyclic loading history on the stiffness and strength of a multifilament coreless UHMWPE surgical suture. The topic is interesting, but I think this article should reconsider after proper changes in major revision for publication in Materials. Some of my specific comments are below:
- Since UHMWPE is a widely adopted biomaterials used in the bearing of total hip arthroplasty. I would encourage and advise the authors to adopt some of the additional related references published by MDPI in the introduction section:
-
- Tresca Stress Simulation of Metal-on-Metal Total Hip Arthroplasty during Normal Walking Activity. Materials (Basel). 2021, 14, 7554. https://doi.org/10.3390/ma14247554
- The Effect of Bottom Profile Dimples on the Femoral Head on Wear in Metal-on-Metal Total Hip Arthroplasty. J. Funct. Biomater. 2021, 12, 38. https://doi.org/10.3390/jfb12020038
- I see some errors on English in some areas of the present manuscript. To improve the quality of English used in this manuscript and make sure English language, grammar, punctuation, spelling, and overall style are correct, further proofreading is needed. As an alternative, the authors can use the MDPI English proofreading service for this issue.
- In the title of the manuscript (line 1-2), the authors are using uncorrected uppercase and lowercase that should be corrected.
- In line 13 of abstract section, the authors state “on the stiffness and strength of a multifilament coreless UHMWPE surgical suture”. Before mentioning UHMWPE, the authors state the stands for the abbreviation UHMWPE before mentioning it.
- In line 15 of abstract section, the authors state “…history and after 10 cycles of loading at [1-10], [1-30] and [1-50]N”. When the ranged nominal is written down like [1-10] or [1-30] or [1-50] as presented in the author's manuscript it will be leading to misperception to the readers. Because [1-10] of looks like citing the references. The authors are advised to revise it to become “1-10N, 1-30N, and 1-50N” or “1N - 10N, 1N - 30N, and 1N - 50N”.
- Describe the novelty of the article made by the author? From the results of my evaluation, it seems that many similar published works adequately explain what you have raised in the current manuscript. As the best reviewer knowledge in this research area, Mechanical properties of sutures have been studied in materials. However, only evaluating characterization of the mechanical properties of UHMWPE suture threads under different loading conditions as stated by authors in line 62-63 is not enough novelty to raisin in the author’s present article. If there is something others really new in this manuscript, please highlight it more clearly in the introduction section (line 28-690.
- The state of the art and the significance of the present study are not clearly present, the authors should highlight it more advanced in the introduction section (line 28-69).
- In the introduction section (line 28-69), the authors should explain the previous research conducted and its shortcomings. It will uphold the research gap that you filled with your research novelty. I recommend the authors elaborate their introduction section.
- Figure 1 (line 80) should be improved by giving part description in the figure rather than only number and explain the figure name.
- The author must provide a detailed specification and use condition more detail regarding all tools used in the research carried out so that the reader can estimate the accuracy and differences in the results that the authors describe due to the use of different tools in future studies.
- In the results section (line 169-302), authors are advised to compare the results they obtain with previous similar/identical studies if it is possible.
- The conclusion (line 428-438) of the present manuscript is not solid. Further elaboration is needed. Also, it suggests elaborating only one comprehensive paragraph.
- Further research needs to be explained in the conclusion section (line 428-438).
- In the whole of manuscript, the authors seem not to put commas (,) for the sentence that mention three or more items, like in line 23 of the abstract section, “the stiffness of the studied suture was influenced by the strain level, loading velocity and recent….”, where it should put after “velocity” and before commas to become “the stiffness of the studied suture was influenced by the strain level, loading velocity, and recent….”. Please correct it.
- Figure A3 (line 523) should be improved by giving zoom in detail for Ecyclic (MPA) data for < 500 cycle. Because the trend in the present figure is not clearly present.
- Figure A4 (line 528) should be improved by giving the zoom in figure for variation 101-1000, 1001-2000, 20001-3000, 3001-4000, 4001-5000, 5001-6000,and 6001-7000. Because the comparison by the figure for this variation is not clearly seen and only seen by exact value as present form.
- Please make sure the authors have used the Materials, MDPI format correctly. The authors can download published manuscripts by Materials, MDPI, and compare them with the present author's manuscript to ensure typesetting is appropriate.
I am pleased to have been able to review the author's present manuscript. Hopefully, the author can revise the current manuscript as well as possible so that it becomes even better. Good luck for the author's work and effort.
Best regards,
The Reviewer
Reviewer 2 Report
Comments on Manuscript ID: materials-1640517, of the title “Loading conditions affect the mechanical performance of multifilament coreless UHMWPE sutures used in orthopaedic surgery”. I have read in detail the manuscript and consider that the subject of this work is interesting, the article addresses the influence of loading velocity and previous cyclic loading history on the stiffness and strength of a multifilament coreless UHMWPE surgical suture and this could be the novelty of this work. The authors have developed a relatively straightforward model to compute the stress-strain relation and the strength of the suture and have demonstrated its validity with the experimental data. The introduction is well written and the work is comprehensive and generally well structured, the discussion is reasonable. However, the following points should be considered for manuscript revision.
Point 1. I recommend authors to rename the paper title.
Point 2. Abstract. the stiffness of the studied suture was influenced by the strain level, loading velocity and recent cyclic loading history. Conversely, the suture strength was not affected. What does this conclusion imply? Avoid generalizing.
Point 3. Line 55-56. Generally, such parameters, tensile strength and stiffness, derive from load-to-failure tests conducted at a single loading velocity [4–6]. Can the authors briefly explain/argue why they are usually only done at a single speed?
Point 4. Materials and methods. Molar mass , molar mass distribution and other important characteristics for the base polymer that the authors used in this study should be provided.
Point 5. According to the USP [9]. USP stands for United States Pharmacopeial Convention?
Point 6. A custom uniaxial traction/compression testing machine was used ... model, country, and so should be mentioned.
Point 7. Should references be added to the models? p11, p21, p12, and p22, ?10 …please define these terms.
Point 8. As for suture strength, no noticeable impact attributable to loading speed was found in the ultimate stress or ultimate stretch ratio. Other authors report this observation? Please clarify.
Point 9. The authors should make a more critical or detailed comparison of their work with other literature reports in terms of properties of the materials characterized in this report with those similar materials reported/obtained by other researchers.
Reviewer 3 Report
Dear Authors
In this manuscript, authors investigated the influence of loading velocity and previous cyclic loading history on the stiffness and strength of multifilament coreless UHMWPE surgical suture.
They concluded that the stiffness of the studied suture was influenced by strain level, loading velocity and recent cyclic loading history. Conversely, the suture strength was not affected.
I have several comments for the authors.
1, For clinical use, sutures can absorb water. What do you think of this data about that?
2, Please make sure there were no typo and grammar errors.(ex.Page10 304,328”UPHWE”→“UHMWPE”)
3, In reference section, it does not comply with the Journal submission rules, please correct it.
I hope these comments will be helpful for improvement of the manuscript.
Best regars
Round 2
Reviewer 1 Report
Dear Prado-Novoa et al.,
After carefully reading the author's revised manuscript entitled "Loading conditions affect the mechanical performance of multifilament coreless UHMWPE sutures used in orthopaedic surgery" (materials-1640517) by Prado-Novoa et al., The authors have been made significant improvements in the revised manuscript. Also, all of the issue in my review report has been addressed precisely.
With my pleasure, I recommend the manuscript should be accepted for publication on Journal of Functional Biomaterials.
Best regards,
The Reviewer